# ACTIVE SPEECH ENHANCEMENT: BEYOND PASSIVE DENOISING DECLIPPING AND DEREVERBERATION

## ABSTRACT

We introduce a new paradigm for active sound modification: *Active Speech Enhancement* (ASE). While Active Noise Cancellation (ANC) algorithms focus on suppressing external interference and traditional speech enhancement passively reconstructs degraded speech signals, ASE goes further by actively shaping the speech signal, both attenuating unwanted noise components and amplifying speech-relevant frequencies to improve intelligibility and perceptual quality. To enable this, we propose a novel Transformer-Mamba-based architecture, along with a task-specific loss function designed to jointly optimize interference suppression and signal enrichment in an acoustic environment. Our method outperforms existing baselines across multiple speech processing tasks, including denoising, dereverberation, and declipping, demonstrating the effectiveness of active, targeted modulation in challenging acoustic environments. A demo page and source code are provided in the Supplementary Materials.

## 1 INTRODUCTION

Traditional speech enhancement and noise control are fundamental audio processing tasks. Traditional speech enhancement aims to passively improve the perceptual quality and intelligibility of speech signals by mitigating degradations such as background noise, distortion, clipping, and reverberation. Classic approaches—including spectral subtraction, Wiener filtering, and statistical model–based methods—have achieved varying degrees of success but often falter in highly non-stationary noise environments (Boll, 2003; Lim & Oppenheim, 1978; Paliwal et al., 2012). Recent advances in deep learning have, however, yielded state-of-the-art performance: convolutional neural networks (CNNs) (Pascual et al., 2017; Rethage et al., 2018; Pandey & Wang, 2018), recurrent neural networks (RNNs) (Hu et al., 2020), generative adversarial networks (GANs) (Fu et al., 2019; 2021; Kim et al., 2021; Shin et al., 2023; Shetu et al., 2025), Transformers (Wang et al., 2021; de Oliveira et al., 2022; Zhang et al., 2022b; Cao et al., 2022; Ye & Wan, 2023; Zhang et al., 2024), and diffusion models (Guimarães et al., 2025; Lu et al., 2022; Welker et al., 2022; Richter et al., 2023; Lemercier et al., 2023; Tai et al., 2023; Ayilo et al., 2024) demonstrate exceptional results on benchmarks for denoising, dereverberation, and declipping.

Active noise cancellation takes a complementary approach by generating an anti-noise signal to interfere with unwanted noise destructively. Pioneering work dating back to Lueg's first patent in 1936 introduced the concept of adaptive feedforward ANC (Lueg, 1936), which was later refined through advances in adaptive filtering (e.g., LMS, FxLMS), multi-channel algorithms, and applications in headphones and enclosure systems (Nelson & Elliott, 1991; Fuller et al., 1996; Hansen et al., 1997; Kuo & Morgan, 1999; Zhang & Wang, 2021; Park et al., 2023; Mostafavi & Cha, 2023; Cha et al., 2023; Singh et al., 2024; Pike & Cheer, 2023; Mishaly et al., 2025). While ANC excels at suppressing predictable or narrowband noise, it does not actively modify the speech content itself.

We propose a new paradigm—*Active Speech Enhancement* (ASE) that unifies the goals of traditional speech enhancement and active noise control. Unlike conventional ANC, which solely targets noise suppression, ASE actively shapes the speech signal by simultaneously attenuating interfering components and amplifying speech-related frequencies. This dual-action approach not only reduces noise but also emphasizes speech and improves perceptual quality under challenging acoustic conditions. We make four key contributions. First, we formalize the ASE task and describe appropriate evaluation metrics that capture both noise suppression and speech enhancement. Second, we

introduce a Transformer-Mamba-based model that generates an active modification signal, leveraging self-attention to capture long-range dependencies in time–frequency representations. Third, we design a joint suppression–enrichment loss that balances interference removal and signal enrichment, combining spectral, perceptual, and adversarial objectives to drive optimal ASE performance. Fourth, we conduct a comprehensive evaluation, showing that our method outperforms adapted baselines across multiple ASE tasks—including denoising, dereverberation, and declipping—with significant gains in metrics such as PESQ (Rix et al., 2001).

## 2 RELATED WORK

### 2.1 TRADITIONAL SPEECH ENHANCEMENT

Recent advances in deep learning have yielded substantial improvements in traditional speech enhancement. Pandey and Wang (Pandey & Wang, 2018) proposed a CNN–based autoencoder that applies convolutions directly to the raw waveform while computing the loss in the frequency domain. SEGAN (Pascual et al., 2017) employs strided convolutional layers in a generative adversarial framework. Rethage et al. (2018) developed a WaveNet-inspired model that predicts multiple waveform samples per step to reduce computational cost. Recurrent architectures have also been explored. Hu et al. (2020) presented DCCRN, which integrates complex-valued convolutional and recurrent layers to process spectrogram inputs. A real-time causal model based on an encoder–decoder with skip connections was proposed by Defossez et al. (2020), operating directly on the raw waveform and optimized in both time and frequency domains.

MetricGAN (Fu et al., 2019) and its successor MetricGAN+ (Fu et al., 2021) incorporate evaluation metrics such as PESQ (Perceptual Evaluation of Speech Quality) (Rix et al., 2001) and STOI (Short-Time Objective Intelligibility) (Kim et al., 2021) into the adversarial loss. Kim et al. (2021) further enhance this approach by introducing a multiscale discriminator operating at different sampling rates alongside a generator that processes speech at multiple resolutions.

Transformer-based models have recently gained prominence. Wang et al. (2021) proposed a two-stage transformer network (TSTNN) that outperforms earlier time- and frequency-domain methods. CMGAN (Cao et al., 2022) adapts the Conformer backbone (Gulati et al., 2020) for enhancement, and de Oliveira et al. (2022) replace the learned encoder of SepFormer (Subakan et al., 2021) with long-frame STFT (Short-Time Fourier Transform) inputs, reducing sequence length and lowering computational cost without compromising perceptual quality.

More recently, diffusion-based approaches have emerged as a powerful generative paradigm. Lu et al. (2022) introduced a conditional diffusion probabilistic model that learns a parameterized reverse diffusion process conditioned on the noisy input. Welker et al. (2022) extended score-based models to the complex STFT domain, learning the gradient of the log-density of clean speech coefficients. Richter et al. (2023) formulate enhancement as a stochastic differential equation, initializing reverse diffusion from a mixture of noisy speech and Gaussian noise and achieving high-quality reconstructions in only 30 steps. Lemercier et al. (2023) propose a *stochastic regeneration* method that leverages an initial predictive-model estimate to guide a reduced-step diffusion process, mitigating artifacts and reducing computational cost by an order of magnitude while maintaining quality.

### 2.2 ACTIVE AUDIO CANCELLATION

Recently, deep learning approaches have demonstrated remarkable results in ANC algorithms. Zhang & Wang (2021) introduced DeepANC, which employs a convolutional long short-term memory (Conv-LSTM) network to jointly estimate amplitude and phase responses from microphone signals. Subsequently, attention-driven ANC frameworks integrating attentive recurrent networks were proposed to enable real-time adaptation and low-latency operation (Zhang et al., 2022a).

A selective fixed-filter ANC (SFANC) framework was developed to leverage a two-dimensional CNN for optimal control-filter selection on a mobile co-processor and a lightweight one-dimensional CNN for time-domain noise classification, yielding superior attenuation of real-world non-stationary headphone noise (Shi et al., 2022). Luo et al. (2022) proposed a hybrid SFANC–FxNLMS that first applies a similar approach as SFANC for each noise frame and then applies the FxNLMS algorithm for real-time coefficient adaptation, thereby combining the rapid response of SFANC with the low

steady-state error and robustness of adaptive optimization. Heuristic algorithms—such as bee colony optimization (Ren & Zhang, 2022) and genetic algorithms (Zhou et al., 2023)—have been explored to avoid gradient-based learning.

Other studies have applied recurrent convolutional networks (Park et al., 2023; Mostafavi & Cha, 2023; Cha et al., 2023) and fully connected neural networks (Pike & Cheer, 2023) to ANC. Autoencoder-based encodings have been used to extract latent features for improved robustness (Singh et al., 2024). Efforts in SFANC have extended to synthesizing optimized filter banks via unsupervised methods (Luo et al., 2024), while advancements in multichannel setups continue to leverage spatial diversity through deep controllers (Shi et al., 2024). Multichannel configurations have been further enhanced by refined deep controllers that learn inter-channel relationships for improved noise attenuation (Zhang & Wang, 2023; Antoñanzas et al., 2023; Xiao et al., 2023; Zhang et al., 2023b; Shi et al., 2023), and attention-driven frameworks have been investigated for low-latency operation (Zhang et al., 2023a).

## 3 BACKGROUND

We first examine a feedforward ANC algorithm that employs a single error microphone to lay the foundation for our new ASE framework. In the ANC framework (Figure 1a), the *primary path*, characterized by the transfer function $P(z)$, models the acoustic propagation from the disturbance source to the error microphone. The *secondary path*, denoted by $S(z)$, describes the transfer from the loudspeaker to the error microphone. Let $x(n)$ denote the reference signal applied to the ANC system. The primary signal $d(n)$ is obtained by filtering $x(n)$ through the primary path:

$$d(n) \ = \ P(z) * x(n) \,, \tag{1}$$

where $*$ denotes the convolution operation. The error microphone captures the residual signal $e(n)$, representing the difference between the original disturbance and the cancellation signal. Both $x(n)$ and $e(n)$ are used by the ANC algorithm to compute the canceling signal $y(n)$. The loudspeaker implements $y(n)$ according to its electro-acoustic transfer function $f_{\mathrm{LS}}\{\cdot\}$. After propagation through the secondary path, the anti-signal (or cancellation signal) is given by

$$a(n) \ = \ S(z) * f_{\mathrm{LS}}\big(y(n)\big) \,. \tag{2}$$

The error signal is defined formally as the difference between the primary signal and the anti-signal:

$$e(n) \ = \ d(n) - a(n) \,. \tag{3}$$

The objective of the ANC algorithm is to generate $y(n)$ such that $e(n)$ is minimized, ideally achieving $e(n) = 0$, which corresponds to complete cancellation of the disturbance. In contrast, the **ASE framework** uses the primary and anti-signals to construct an enhanced signal:

$$eh(n) \ = \ d(n) + a(n) \,. \tag{4}$$

While ANC aims to eliminate the disturbance, ASE seeks to recover clean speech from a noisy mixture of distorted speech $x(n)$. The objective of the ASE task is to generate $eh(n)$ such that its deviation from the clean target signal $c(n)$, i.e., $eh(n) - c(n)$, is minimized. As illustrated in Figure 1b, the feedforward ASE setup comprises a disturbance source, a reference signal path, and a control filter operating through the secondary path. Given the nature of the task, the error microphone serves as the modification microphone in our framework.

Our work adapts three speech distortion types, previously defined by VoiceFixer (Liu et al., 2022) for general speech restoration, to the context of our ASE framework. Specifically, our ASE-TM model targets the restoration of speech $s(n)$ degraded by: **(i) Additive noise**: This common distortion, where unwanted background sounds obscure the speech, is modeled as the sum of the clean speech signal $s(n)$ and a noise signal $n(n)$:

$$d_{\mathrm{noise}}(s(n)) = s(n) + n(n) \,. \tag{5}$$

**(ii) Reverberation**: Caused by sound reflections in an enclosure, reverberation blurs speech signals. It is modeled by convolving the speech signal $s(n)$ with a room impulse response (RIR) $r(n)$:

$$d_{\mathrm{rev}}(s(n)) = s(n) * r(n) \,. \tag{6}$$

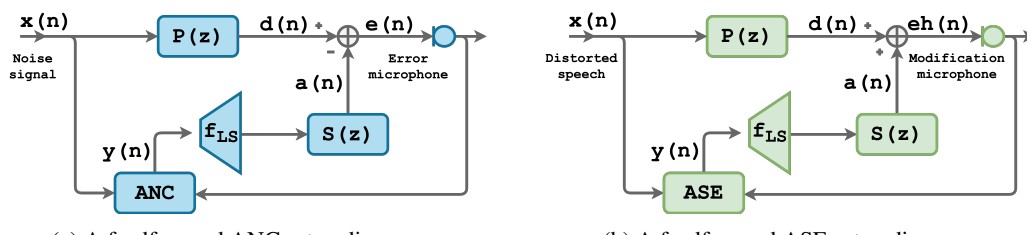

(a) A feedforward ANC setup diagram.  (b) A feedforward ASE setup diagram.

Figure 1: Comparison of feedforward ANC and ASE setups.

**(iii) Clipping**: This distortion arises when signal amplitudes exceed the maximum recordable level, typically due to microphone limitations. Clipping truncates the signal $s(n)$ within a certain range $[-\eta, +\eta]$:

$$d_{\text{clip}}(s(n)) = \max(\min(s(n), \eta), -\eta), \quad \eta \in [0, 1].  \tag{7}$$

This leads to harmonic distortions and can degrade speech intelligibility.

To assess the performance of ASE-TM across these enhancement tasks, we employ a suite of established objective metrics. Consistent with the evaluation protocol in SEmamba (Chao et al., 2024), these include the Wide-band PESQ (Rix et al., 2001), STOI (Taal et al., 2010), and the composite measures CSIG (predicting signal distortion), CBAK (predicting background intrusiveness), and COVL (predicting overall speech quality) (Hu & Loizou, 2007). Furthermore, we incorporate the Normalized Mean Square Error (NMSE), a traditionally well-established metric in the ANC task. The NMSE between a target signal $u(n)$ and an estimated signal $v(n)$ is defined in decibels (dB) as:

$$\text{NMSE}[u, v] = 10 \cdot \log_{10} \left( \frac{\sum_{n=1}^{M} (u(n) - v(n))^2}{\sum_{n=1}^{M} u(n)^2} \right),  \tag{8}$$

where $M$ is the total number of samples. In our evaluations, $u(n)$ represents the clean target speech $c(n)$, and $v(n)$ is the enhanced speech $eh(n)$ produced by our model (the precise definition of $c(n)$ for each task is detailed in Section 4.1).

## 4 METHOD

### 4.1 ASE-TM ARCHITECTURE OVERVIEW

The proposed model, ASE Transformer-Mamba (ASE-TM, Figure 2), adopts and extends the fundamental structure of the SEmamba architecture (Chao et al., 2024), which consists of a dense encoder, a series of Time-Frequency Mamba (TFMamba) blocks (Xiao & Das, 2024), and parallel magnitude and phase decoders. A notable distinction of our ASE-TM model is the utilization of Mamba2 blocks (Dao & Gu, 2024) within these TFMamba pathways, in contrast to the original SEmamba architecture, which employed an earlier version of Mamba (Gu & Dao, 2023). This choice is motivated by the potential improvements in efficiency offered by Mamba2.

The input noisy waveform, $x(n)$, sampled at a rate of $F_s$, is processed through an STFT. The STFT employs a Hann window of $N_{\text{win}}$ samples, a hop length of $N_{\text{hop}}$ samples, and an $N_{\text{FFT}}$-point FFT, resulting in $N_{\text{freq}} = \lfloor N_{\text{win}}/2 \rfloor + 1$ frequency features per frame. Its magnitude and phase components are horizontally stacked and fed into the network. The dense encoder utilizes convolutional layers and dense blocks to extract initial features from the stacked magnitude and phase, outputting a representation with $C_{\text{enc}}$ channels, each with $N_{\text{enc}}$ features.

The core of the temporal and spectral modeling is based on $N_{\text{tf}}$ TFMamba blocks. Each TFMamba block contains separate Mamba-based pathways (`time-mamba` and `freq-mamba`) employing bidirectional Mamba layers to capture dependencies across time and frequency dimensions, respectively (Chao et al., 2024; Xiao & Das, 2024).

Following the initial $N_{\text{tf}}/2$ TFMamba blocks, inspired by hybrid approaches like Jamba (Lieber et al., 2024), we introduce an attention-based block. Before applying attention, the feature representation, with $C_{\text{enc}}$ channels, undergoes dimensionality reduction. A 2D convolution with a kernel size

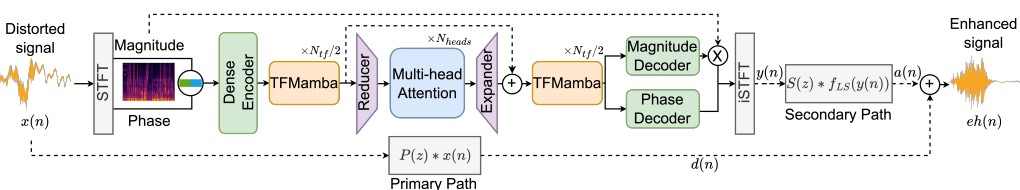

Figure 2: ASE-TM Architecture.

of $(1, N_{enc}/2 + 1)$ reduces the channel dimension from $C_{enc}$ to $C_{enc}/4$ and each channel features size from $N_{enc}$ to $N_{enc}/2$. This results in a compact representation of size $N_{enc}/2 \times C_{enc}/4$ for the attention layer. In addition, positional encoding is applied to this compact representation. A standard Multi-Head Attention layer with $N_{heads}$ heads is then used on this reduced representation to weigh features based on global context. Following the attention layer, an expansion module employing a transposed 2D convolution, also with a kernel size of $(1, N_{enc}/2 + 1)$, is used to restore the channel dimension to $C_{enc}$ and expand the feature dimension back towards $N_{enc}$ before passing the features to the remaining $N_{tf}/2$ TFMamba blocks. An additional step before applying the remaining TF-Mamba blocks is the use of a residual connection that sums the feature representations from before the dimensionality reduction with those after the attention and expansion modules

The magnitude and phase decoders retain the structure used in SEmamba, employing dense blocks and convolutional layers (including transposed convolutions) to reconstruct the target representation—not before applying a residual connection that performs element-wise multiplication between the original STFT magnitude and the predicted magnitude. However, instead of predicting the enhanced spectra directly, the network is trained to output the complex spectrum of the cancelling signal $y(n)$. This signal $y(n)$, after undergoing the electro-acoustic transfer function $f_{LS}\{\cdot\}$ and propagation through the secondary path $S(z)$, becomes the anti-signal $a(n)$ (as defined in Eq. 2). This anti-signal $a(n)$ is then summed with the primary path signal $d(n)$ to produce the final enhanced signal $eh(n)$ (as defined in Eq. 3).

## 4.2 OPTIMIZATION OBJECTIVE

The primary goal of the ASE-TM model is to generate an enhanced signal $eh(n)$ that is as close as possible to a clean target speech signal $c(n)$. The definition of this target $c(n)$ varies based on the specific enhancement task. For **additive noise reduction**, $c(n)$ is the clean speech signal convolved with the primary path $P(z)$, representing the clean signal as perceived at the modification microphone. For **dereverberation** and **declipping**, $c(n)$ is the original anechoic, unclipped clean speech signal, prior to any acoustic path effects or clipping distortion.

The training of ASE-TM largely follows the multi-level loss framework established in SEmamba and originating from MP-SENet (Lu et al., 2023). This framework combines several loss components. Our approach incorporates this established framework with specific modifications. The overall generator loss $\mathcal{L}_G$ is a weighted sum of the following components:

1. **Time-Domain Loss ($\mathcal{L}_T$)**: We employ a combination of L1 and L2 distances between the enhanced waveform $eh(n)$ and the target waveform $c(n)$:

$$\mathcal{L}_T = ||eh(n) - c(n)||_1 + ||eh(n) - c(n)||_2^2. \tag{9}$$

   This hybrid loss aims to leverage the robustness of L1 to outliers and the smoothness encouraged by L2.

2. **Magnitude Spectrum Loss ($\mathcal{L}_{Mag}$)**: Similar to the time-domain loss, a combined L1 and L2 loss on the magnitude spectra is applied. If $EN_m$ and $C_m$ are the magnitude spectra of $eh(n)$ and $c(n)$ respectively, then:

$$\mathcal{L}_{Mag} = ||EN_m - C_m||_1 + ||EN_m - C_m||_2^2. \tag{10}$$

   This contrasts with the L2 loss typically used in MP-SENet for this component.

3. **Complex Spectrum Loss ($\mathcal{L}_{Com}$)**: This loss penalizes differences in the STFT domain. It is the sum of L2 losses on the real and imaginary parts of the STFTs of $eh(n)$ and $c(n)$.

4. **Anti-Wrapping Phase Loss ($\mathcal{L}_{\mathbf{Pha}}$)**: Includes instantaneous phase loss, group delay loss, and instantaneous angular frequency loss to optimize the phase spectrum directly, addressing phase wrapping issues.

5. **Metric-Based Adversarial Loss ($\mathcal{L}_{\mathbf{Met}}$)**: A discriminator trained to predict a perceptual metric (e.g., PESQ), guiding the generator to produce outputs that score well on it.

6. **Consistency Loss ($\mathcal{L}_{\mathbf{Con}}$)**: We incorporate a consistency loss. This loss minimizes the discrepancy between the complex spectrum directly output by the model's decoders (magnitude and phase) and the complex spectrum obtained by applying STFT to the time-domain waveform $eh(n)$ that results from an inverse STFT of the initially predicted spectrum.

The total generator loss is then defined with the hyperparameter $\gamma$ as follows:

$$\mathcal{L}_G = \gamma_1 \mathcal{L}_{\text{T}} + \gamma_2 \mathcal{L}_{\text{Mag}} + \gamma_3 \mathcal{L}_{\text{Com}} + \gamma_4 \mathcal{L}_{\text{Met}} + \gamma_5 \mathcal{L}_{\text{Pha}} + \gamma_6 \mathcal{L}_{\text{Con}}. \tag{11}$$

## 5 EXPERIMENTS

### 5.1 DATASETS AND TASK GENERATION

We conduct evaluations across three primary speech restoration tasks: additive noise reduction, dereverberation, and declipping. For **additive noise reduction**, we use the VoiceBank-DEMAND dataset (Botinhao et al., 2016), a standard benchmark in speech enhancement. This dataset combines clean speech from the VoiceBank corpus (Veaux et al., 2013) with various non-stationary noises from the DEMAND database (Thiemann et al., 2013). Our training set consists of utterances from 28 speakers with 10 different noise types at Signal-to-Noise Ratios (SNRs) of 0, 5, 10, and 15 dB. We used two speakers from the training set as the validation set. The test set comprises 824 utterances from 2 unseen speakers, mixed with 5 unseen noise types at SNRs of 2.5, 7.5, 12.5, and 17.5 dB.

The datasets for the **dereverberation** and **declipping** tasks are generated using the clean speech utterances of the speakers available in the VoiceBank corpus. **Dereverberation**: Reverberant speech is synthesized by convolving the clean VoiceBank utterances with Room Impulse Responses (RIRs) as defined in Eq. 6 using the SpeechBrain package (Ravanelli et al., 2024). For training, we randomly sample RIRs from the training portion of the RIR dataset provided alongside VoiceFixer (Liu et al., 2022). For the test set, a fixed and distinct set of RIRs (from the VoiceFixer RIR test set) is applied to the clean test utterances to ensure consistent evaluation conditions. **Declipping**: Clipped speech signals are generated by applying a clipping threshold $\eta$ to the clean utterances according to Eq. 7. During training, the clipping ratio $\eta$ is uniformly sampled from the range $[0.1, 0.5]$ for each utterance to expose the model to varying degrees of distortion. For testing, a specific clipping threshold is used.

### 5.2 ACOUSTIC PATH SIMULATION

To emulate the acoustic environment for the ASE framework, we simulate the primary path $P(z)$ and secondary path $S(z)$. Our simulation setup is based on previous setups for the ANC task (Zhang & Wang, 2021; Zhang et al., 2023a), modeling a rectangular enclosure with dimensions of $3 \times 4 \times 2$ meters (width × length × height). Room Impulse Responses (RIRs) are generated using the image method (Allen & Berkley, 1979), implemented with a Python-based RIR generator package (Habets, 2006) with high-pass filtering. The modification microphone, capturing $eh(n)$, is at $[1.5, 3, 1]$ meters. The reference microphone, capturing $x(n)$, is at $[1.5, 1, 1]$ meters, and the cancellation loudspeaker, which outputs the signal leading to $a(n)$, is at $[1.5, 2.5, 1]$ meters within the enclosure. The RIR length for both $P(z)$ and $S(z)$ is $L_{\text{RIR}} = 512$ taps. The non-linear characteristics of the loudspeaker are modeled using the Scaled Error Function (SEF), defined as $f_{\text{LS}}\{y\} = \int_0^y \exp(-z^2/(2\lambda^2)) dz$. Here, $y$ is the loudspeaker input, and $\lambda^2$ controls the severity of the saturation non-linearity. Different $\lambda^2$ values simulate varying degrees of distortion, with larger values approaching linear behavior. To introduce variability during training, the room's reverberation time ($T_{60}$) and $\lambda^2$ are randomly sampled from the sets $\{0.15, 0.175, 0.2, 0.225, 0.25\}$ seconds and $\{0.1, 1, 10, \infty\}$, respectivly, for each training sample. For testing, fixed $T_{60}$ and $\lambda^2$ are used.

Table 1: Average denoising results on the VoiceBank-DEMAND test set ($T_{60} = 0.25s$ and $\lambda^2 = \infty$).

| Method | PESQ (↑) | CSIG (↑) | CBAK (↑) | COVL (↑) | STOI (↑) | NMSE (↓) |
|--------|----------|----------|----------|----------|----------|----------|
| Noisy-speech | 1.97 | 3.50 | 2.55 | 2.75 | 0.92 | -8.44 |
| THF-FxLMS | 2.37 | 3.66 | 2.84 | 3.00 | 0.97 | -15.32 |
| DeepANC | 1.48 | 1.99 | 2.19 | 1.69 | 0.93 | -12.80 |
| ARN | 2.45 | 3.64 | 3.13 | 3.03 | 0.97 | -20.64 |
| ASE-TM | **2.98** | **4.21** | **3.49** | **3.62** | **0.99** | **-21.76** |

## 5.3 MODEL HYPERPARAMETERS AND TRAINING

The ASE-TM model processes audio at $F_s = 16$ kHz. For the STFT, we use a Hann window of $N_{\text{win}} = 400$ samples, hop length of $N_{\text{hop}} = 100$ samples, and an $N_{\text{FFT}} = 400$-point FFT. The dense encoder outputs a feature representation with $C_{\text{enc}} = 128$ channels, where each channel has a feature dimension of $N_{\text{enc}} = 100$. Our model employs a total of $N_{\text{tf}} = 8$ TFMamba blocks. The Multi-Head Attention layer within the attention-based block uses $N_{\text{heads}} = 10$ heads. Other internal architectural details for the TFMamba blocks and dense convolutional blocks largely follow the configurations presented in SEmamba. The ASE-TM model is trained for 350 epochs using the AdamW optimizer (Loshchilov & Hutter, 2017) with $\beta_1 = 0.8$ and $\beta_2 = 0.99$. The initial learning rate is set to $5 \times 10^{-4}$. We use a batch size of 4. Audio segments of $32,000$ samples (equivalent to 2 seconds at 16 kHz) are used for training. The model parameters yielding the best performance on the validation set, evaluated based on the PESQ score, are saved for final testing. We used an NVIDIA RTX A6000 GPU (internal cluster). The training runtime of the ASE-TM model was $\sim 10$ days.

## 5.4 BASELINE METHODS

We compare ASE-TM with several established baseline methods commonly used in ANC. These include THF-FxLMS (Ghasemi et al., 2016), which is an extension to the traditional FxLMS algorithm (Kuo & Morgan, 1999), DeepANC that utilizes a convolutional LSTMs (Zhang & Wang, 2021), and ARN that incorporates an attention mechanism (Zhang et al., 2023a). These baseline methods were adapted and retrained or configured to the ASE framework across all tested tasks.

# 6 RESULTS AND ANALYSIS

## 6.1 ACTIVE DENOISING PERFORMANCE

The speech denoising performance on the VoiceBank-DEMAND dataset is in Table 1. Our ASE-TM model demonstrates superior performance, achieving a PESQ score of 2.98, significantly surpassing the baselines: THF-FxLMS achieved a PESQ of 2.37, and the deep learning-based ANC methods, DeepANC and ARN, yielded PESQ scores of 1.48 and 2.45, respectively. These results demonstrate a substantial gap between conventional ANC approaches and our ASE-TM, which benefits from actively shaping the speech signal in addition to noise suppression, as also evidenced by its leading scores in CSIG, CBAK, COVL, STOI, and a significantly better NMSE of $-21.76$ dB.

## 6.2 DEREVERBERATION AND DECLIPPING PERFORMANCE

ASE-TM's efficacy was further evaluated on dereverberation and declipping tasks, with results presented in Tables 2 and 3, respectively. For dereverberation (Table 2), ASE-TM achieved a PESQ score of 2.43, a considerable improvement from the reverberant speech baseline (PESQ 1.60). In contrast, the adapted ANC baselines struggled; THF-FxLMS scored a PESQ of 1.43, while Deep-ANC and ARN achieved 1.06 and 1.35, respectively. This suggests that these methods, even when retrained or configured for the task, struggled to effectively adjust their processes to mitigate reverberation in the ASE framework. Similarly, in the declipping task, with a clipping threshold of $\eta = 0.25$ (Table 3), ASE-TM restored speech to a PESQ of 3.09 from an initial score of 2.17. The baseline methods again showed limited effectiveness: THF-FxLMS (PESQ 1.92), DeepANC (PESQ 1.05), and ARN (PESQ 1.67). These tasks, particularly where the target $c(n)$ is the original clean speech before any primary path effects, highlight the challenge and efficacy of the ASE approach in not just cancelling an interfering signal but actively restoring a desired signal characteristic.

Table 2: Average dereverberation results on the reverbed test set ($T_{60} = 0.25s$ and $\lambda^2 = \infty$).

| Method | PESQ (↑) | CSIG (↑) | CBAK (↑) | COVL (↑) | STOI (↑) | NMSE (↓) |
|---|---|---|---|---|---|---|
| Reverbed-speech | 1.60 | 2.60 | 1.88 | 2.02 | 0.80 | 2.00 |
| THF-FxLMS | 1.43 | 2.55 | 1.64 | 1.89 | 0.78 | 4.77 |
| DeepANC | 1.06 | 1.00 | 1.00 | 1.00 | 0.53 | 21.19 |
| ARN | 1.35 | 1.25 | 1.54 | 1.18 | 0.76 | 4.58 |
| ASE-TM | **2.43** | **3.71** | **2.67** | **3.07** | **0.93** | **-0.04** |

Table 3: Average declipping results on the clipped test set ($\eta = 0.25$, $T_{60} = 0.25s$, and $\lambda^2 = \infty$).

| Method | PESQ (↑) | CSIG (↑) | CBAK (↑) | COVL (↑) | STOI (↑) | NMSE (↓) |
|---|---|---|---|---|---|---|
| Clipped-speech | 2.17 | 3.49 | 2.51 | 2.82 | 0.89 | -0.23 |
| THF-FxLMS | 1.92 | 3.35 | 2.36 | 2.62 | 0.88 | -0.02 |
| DeepANC | 1.05 | 1.00 | 1.00 | 1.00 | 0.53 | 11.10 |
| ARN | 1.67 | 1.60 | 2.12 | 1.57 | 0.87 | -0.31 |
| ASE-TM | **3.09** | **4.20** | **3.06** | **3.67** | **0.93** | **-1.70** |

## 6.3 DENOISING ASE-TM MODEL ANALYSIS

An ablation study, presented in Figure 3a, investigates the contributions of our proposed loss function modifications, the attention mechanism, and the use of Mamba2 over Mamba1 to the ASE-TM model for the denoising task. The full model consistently achieves the highest validation PESQ score throughout training. Replacing Mamba1 with Mamba2 and the modified loss yielded the most considerable performance improvement among all evaluated components. Notably, configurations incorporating the attention mechanism demonstrate a faster convergence to higher performance levels, suggesting that attention aids in efficiently learning relevant features. Spectrogram analysis of a representative denoising example (Figure 3b) visually confirms the model's effectiveness; the spectrogram of the enhanced signal closely mirrors that of the clean speech (after primary path), indicating successful noise suppression while preserving essential speech characteristics.

To assess robustness, ASE-TM was evaluated under varying acoustic conditions for the denoising task, with results in Table 4. This analysis focused on ASE-TM due to its significantly better performance over baselines in Table 1. The model shows consistent high performance across different $T_{60}$ values under linear loudspeaker conditions ($\lambda^2 = \infty$), achieving a PESQ of 3.02 for $T_{60} = 0.15s$ and 3.13 for $T_{60} = 0.20s$. When strong non-nonlinearities are introduced (e.g., $\lambda^2 = 0.1$ at $T_{60} = 0.25s$), the PESQ score is 2.74, still indicating robust performance. As $\lambda^2$ increases (less non-linearity), performance improves, reaching a PESQ of 2.97 for $\lambda^2 = 10$ at $T_{60} = 0.25s$.

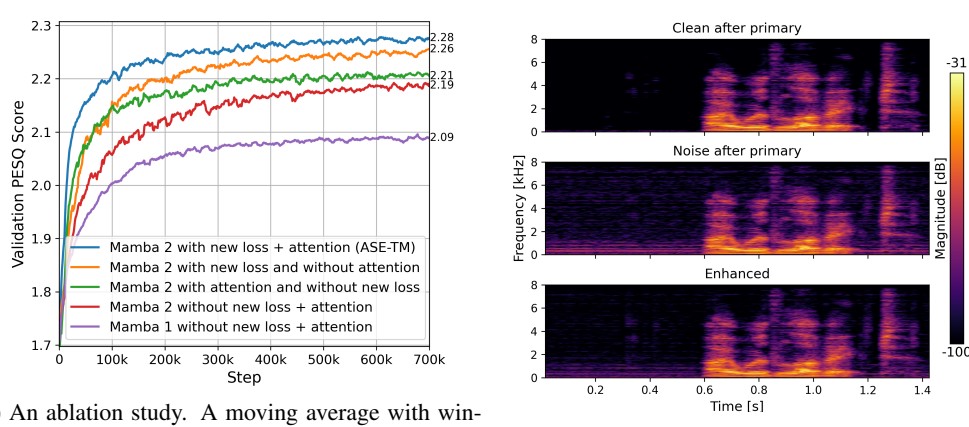

(a) An ablation study. A moving average with window size = 10 was applied.

(b) An enhanced (denoised) signal spectrogram.

Figure 3: Model analysis of ASE-TM model for the denoising task.

Table 4: Average performance of ASE-TM (denoising task) under varying conditions ($T_{60}$ and loudspeaker non-linearity factor $\lambda^2$) on the VoiceBank-DEMAND test set.

| $T_{60}$ (s) | $\lambda^2$ | PESQ (↑) | CSIG (↑) | CBAK (↑) | COVL (↑) | STOI (↑) | NMSE (↓) |
|---|---|---|---|---|---|---|---|
| 0.25 | 0.1 | 2.74 | 4.01 | 3.29 | 3.39 | 0.98 | -20.21 |
| 0.25 | 1.0 | 2.92 | 4.17 | 3.44 | 3.57 | 0.99 | -21.92 |
| 0.25 | 10 | 2.97 | 4.21 | 3.48 | 3.62 | 0.99 | -22.37 |
| 0.15 | $\infty$ | 3.02 | 4.22 | 3.50 | 3.65 | 0.98 | -22.31 |
| 0.20 | $\infty$ | 3.13 | 4.33 | 3.60 | 3.77 | 0.99 | -22.88 |

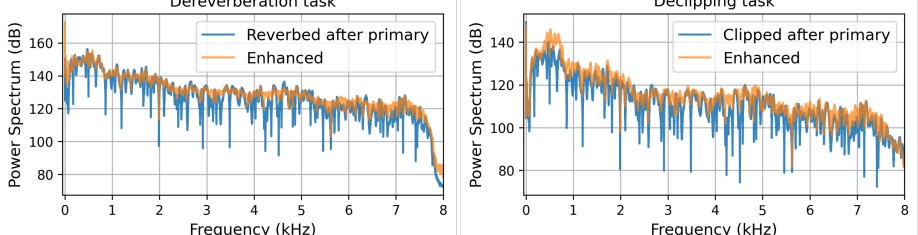

Figure 4: Power spectra for the dereverberation and declipping tasks over the entire test set.

## 6.4 RUNTIME ANALYSIS

To satisfy real-time constraints in active systems, we evaluated ASE-TM under a future-frame prediction strategy, following prior work (Zhang & Wang, 2021; Zhang et al., 2023a). In our setup, the causality condition $T_{\text{ASE-TM}} < T_p - T_s$ evaluates to $T_{\text{ASE-TM}} < \frac{2}{343} - \frac{0.5}{343} \approx 0.0043$ seconds, where $T_p$ and $T_s$ denote the acoustic delays of the primary and secondary paths, respectively. To accommodate the model's inference latency, we predict 500 future frames (0.03125 seconds), remaining within real-time limits of our computational environment. Despite this future context, performance degradation is minimal: on the VoiceBank-DEMAND test set ($T_{60} = 0.25$ s, $\eta^2 = \infty$), ASE-TM achieves a PESQ of 2.96 and STOI of 0.99—closely matching the non-causal configuration.

## 6.5 DEREVERBERATION AND DECLIPPING ASE-TM MODEL ANALYSIS

Figure 4 presents the power spectra of the enhanced signals for the dereverberation and declipping tasks, over the entire test set. For both tasks, the spectrum of the enhanced signal $eh(n)$ exhibits significantly more power across a broad frequency range compared to the distorted input (reverberated or clipped after primary path), indicating successful signal restoration and enrichment. In the declipping task, it is particularly noteworthy that lower frequencies, crucial for speech intelligibility, show substantial power recovery in the enhanced signal's spectrum. We further evaluated the declipping performance under a more aggressive clipping threshold of $\eta = 0.1$. The unprocessed clipped speech at this level yielded a PESQ score of $1.53$ and an NMSE of $-0.18$ dB. ASE-TM restored these signals to a PESQ of $2.52$ (CSIG $3.61$, CBAK $2.76$, COVL $3.08$, STOI $0.91$) and an NMSE of $-1.22$ dB. While these results are lower than for $\eta = 0.25$, they represent a substantial improvement over the severely clipped input, showing ASE-TM's capability to handle extreme distortions.

## 7 CONCLUSIONS AND LIMITATIONS

In this paper, we introduced ASE, a novel paradigm that extends beyond traditional ANC by actively shaping the speech signal to enhance quality and intelligibility. Our ASE-TM model, which leverages a Transformer-Mamba architecture and a specialized loss function, demonstrated strong performance in denoising, dereverberation, and declipping, outperforming baseline methods. This study also reveals limitations that require further investigation. Baseline methods, designed for ANC, were adapted to ASE tasks, which may explain their reduced performance. Furthermore, future work should focus on developing a unified model that can handle multiple speech enhancement objectives, potentially leading to more versatile and efficient systems.

We utilized large language models (LLMs) to assist in refining the manuscript's writing.

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
