# OpenReview forum: "Active speech enhancement: beyond passive denoising declipping and dereverberation"
_ICLR.cc/2026/Conference — ICLR 2026 Conference Withdrawn Submission_

### Official Review · Reviewer_mgHN · 2025-10-27

**Soundness:** 3
**Presentation:** 3
**Contribution:** 2
**Rating:** 2
**Confidence:** 4

**Summary:**

The authors propose using the Active Noise Cancellation (ANC) paradigm for speech enhancement. They use Transformer- and Mamba-based architectures, applied in the spectral domain, to model the secondary path signal. The model's output is regressed against the ground-truth clean signal, and the model parameters are optimised using several regression losses. The model is compared against several baselines on the VoiceBank-DEMAND dataset.

**Strengths:**

I would like to highlight the following strengths that the paper possesses:

1. The paper is well-written and well-structured, which makes it easy to follow and understand the main ideas.
2. The background and the model description are well-presented and clear.
3. The experiment results are well-described.

**Weaknesses:**

However, the paper also has several weaknesses:

1. The model is trained using a linear combination of 6 losses. However, no ablation study is conducted to showcase the importance and the effect of each loss. There is no discussion about the benefits of each loss. A discussion about the methods used to choose the $\gamma_i$ weights in loss functions is also lacking. I believe this discussion is necessary to shed some light on the intricacies of training and provide a clearer picture of how the model behaves.
2. The paper lacks results for the non-intrusive metrics, such as DNSMOS [1], UTMOS [2], or WV-MOS [3], despite their abundant presence in the literature (see, for example, [4 - 6, inter alia]). Moreover, some previous works [3, 7, 8] indicate their benefits over traditional metrics like PESQ. I would recommend including the results for the aforementioned metrics in the paper or justifying why those metrics should not be used for this particular case. Computing the Mean Opinion Score (MOS) would also strengthen the paper.
3. It is worth considering applying the model to other benchmarks, as VoiceBank-DEMAND is a relatively simple benchmark with audio samples containing only mild distortions.
4. Oftentimes, the advantage of the presented model is difficult to perceive while listening to the audio samples given in the supplemental material. Despite outperforming the baselines in PESQ and other metrics, the presented model sounds similar to some of the baselines. Therefore, it is quite hard to appreciate the advantage of the model perceptually. To this end, I would recommend providing samples that clearly showcase the advantage of the presented model, as well as reporting perceptual metrics (MOS, UTMOS, etc) to substantiate the improvements in perceptual quality.

**Questions:**

I have a few questions about the work.

1. It is unclear to me why considering ANC paradigm for speech enhancement is beneficial, while there exist several works [4, 5, 6] that successfully solve the speech enhancement without ANC paradigm.
2. Some applications of speech enhancement include deploying the models on wearable devices and using the models in a streaming fashion. These applications require high inference speed; moreover, it can be beneficial to make the model autoregressive. I wonder what the inference speed of the model is (in terms of RTF), and if the model can be made autoregressive.

References:

[1] Reddy et al., "DNSMOS P. 835: A non-intrusive perceptual objective speech quality metric to evaluate noise suppressors"

[2] Saeki et al., "UTMOS: UTokyo-SaruLab system for VoiceMOS challenge 2022"

[3] Andreev et al., "HiFi++: a unified framework for bandwidth extension and speech enhancement"

[4] Su et al., "HiFi-GAN-2: Studio-quality speech enhancement via generative adversarial networks conditioned on acoustic features."

[5] Babaev et al., "FINALLY: fast and universal speech enhancement with studio-like quality"

[6] Guimarães et al., "High-fidelity generative speech enhancement via latent diffusion Transformers"

[7] Manocha et al., "Audio similarity is unreliable as a proxy for audio quality."

[8] Manjunath T., "Limitations of perceptual evaluation of speech quality on VoIP systems."

---

### Official Review · Reviewer_7zK6 · 2025-10-31

**Soundness:** 1
**Presentation:** 3
**Contribution:** 1
**Rating:** 0
**Confidence:** 4

**Summary:**

This paper proposes a novel paradigm for speech enhancement "Active Speech Enhancement" (ASE) which is inspired by Active Noise Cancellation (ANC).

It proposes also a transformer-mamba architecture for this novel framework which extends SEMamba proposed firstly for speech enhacement.

The method is tested on different datasets and distortion types (denoising, and dereverberation + declipping) including the widely used VoiceBank-DEMAND denoising dataset against different baseline systems taken from ANC literature and shows superior performance.

**Strengths:**

The transformer-mamba architecture seems effective.

**Weaknesses:**

The paper claims a novel approach to ANC called active speech enhancement where the model instead of predicting a cancellation signal predicts an additive signal.
I think the claim of novel paradigm is not significant.

The paper seems to present just a standard supervised speech enhancement approach where the target c(n) = P(z) * s_clean(n) simply has an extra acoustic path simulation.
As such in the paper the authors should probably compare with SotA techniques in speech enhancement (SEMamba, TFGridNet and so on).
But only comparisons with ANC algorithms are made, but these rely on different assumptions.

Moreover, 'These baseline methods were adapted and retrained or configured to the ASE framework across all tested tasks.'
But how these where adapted ?
Looking at the Tables in some cases some models have even worse PESQ (DeepANC in particular) than noisy signal which calls the experimental validation into question or the fairness of this comparison. Have the authors adapted these baselines to the ASE framework (supervised speech enhancement) ? Or kept ANC ?

The better results are given by the powerful transformer-mamba architecture which is likely much stronger than the other baseline ANC models, and also by the use of the many loss components.
This clearly is valuable but I think not enough for this venue and also the claim of new paradigm is unsupported.


Minor:

The NLP keyword is not needed in my opinion.

**Questions:**

Some questions were placed in the weaknesses section.

---

### Official Review · Reviewer_saHn · 2025-11-02

**Soundness:** 2
**Presentation:** 3
**Contribution:** 2
**Rating:** 2
**Confidence:** 5

**Summary:**

This paper introduces a new paradigm called Active Speech Enhancement (ASE), which aims to go beyond traditional passive speech enhancement and active noise cancellation by actively shaping the speech signal. Unlike conventional approaches that only suppress noise or reverberation, ASE both attenuates interference and amplifies speech-relevant frequency components to improve perceptual quality. The authors propose a Transformer–Mamba–based model (ASE-TM) that generates an active modification signal through time–frequency Mamba blocks and an attention mechanism, coupled with a joint suppression–enrichment loss balancing denoising and signal enhancement. Comprehensive experiments across denoising, dereverberation, and declipping tasks show consistent improvements over classical and deep ANC baselines (e.g., DeepANC, ARN, THF-FxLMS) in PESQ, STOI, and NMSE metrics. The results demonstrate ASE-TM’s robustness under diverse acoustic and nonlinear conditions, suggesting its potential as a unified framework for active and intelligent speech restoration.

**Strengths:**

1. Clear writing and decent experimental reproducibility.

2. The idea of linking active control theory with speech enhancement could, in principle, be interesting if more rigorously developed.

3. The adoption of Mamba2 might inspire follow-up studies in sequence modeling for SE.

**Weaknesses:**

1. Conceptual Ambiguity in Task Definition
The paper claims to “formalize the ASE task and describe appropriate evaluation metrics that capture both noise suppression and speech enhancement.” However, in Section 3 (Eq. 4), the formulation directly borrows the notation from Active Noise Control (ANC), where the primary signal d(n) and anti-signal a(n) have clear physical meanings. In the proposed ASE setting, a(n) no longer corresponds to an anti-noise component with opposite phase or amplitude, and there is no identifiable physical counterpart in the presented model. This raises fundamental doubts about whether ASE constitutes a truly independent task, rather than merely a reinterpretation of ANC applied to speech enhancement. The authors should clarify what specific theoretical or practical advantage the ANC framework offers over conventional speech enhancement formulations.

2. Limited Novelty in Model Architecture
The proposed “Transformer–Mamba–based model” is nearly identical to SE-Mamba, with the main change being the replacement of Mamba with Mamba2. While Mamba2 has demonstrated strong modeling ability in ASR [1], its integration here is a straightforward substitution rather than a conceptually new design. The claimed novelty therefore lies more in model adoption than in architectural innovation.

3. Minimal Contribution in Loss Design
The proposed “joint suppression–enrichment loss” combines six components, five of which are identical to those used in MP-SENet. The only addition, L_T, represents a combined time–frequency loss that has already appeared in earlier works (e.g., [2]). Thus, the contribution in objective design is marginal and largely derivative of prior art.

4. Weak Empirical Performance
As shown in Table 1, the denoising results of the proposed method fall significantly short of current state-of-the-art systems [3]. This performance gap undermines the paper’s claim that the ASE formulation or the proposed model yields superior practical benefits.

[1] Y. Masuyama, K. Miyazaki and M. Murata, "Mamba-Based Decoder-Only Approach with Bidirectional Speech Modeling for Speech Recognition," 2024 IEEE Spoken Language Technology Workshop (SLT), Macao, 2024, pp. 1-6, doi: 10.1109/SLT61566.2024.10832186.
[2] Chen, Hang, et al. "Cross-attention among spectrum, waveform and SSL representations with bidirectional knowledge distillation for speech enhancement." Information Fusion (2025): 103218.
[3] M. S. Khan, M. L. Quatra, K. -H. Hung, S. -W. Fu, S. M. Siniscalchi and Y. Tsao, "Exploiting Consistency-Preserving Loss and Perceptual Contrast Stretching to Boost SSL-Based Speech Enhancement," 2024 IEEE 26th International Workshop on Multimedia Signal Processing (MMSP), West Lafayette, IN, USA, 2024, pp. 1-6, doi: 10.1109/MMSP61759.2024.10743615.

**Questions:**

1. What concrete benefits does the ASE formalization provide compared to conventional SE frameworks?

2. What is the physical or algorithmic interpretation of a(n) in your system, given that it no longer functions as an anti-noise signal?

3. Can you isolate the gain of Mamba2 over SE-Mamba through controlled experiments?

4. Have you compared your method with recent diffusion-based or GAN-based SE systems?

---

### Note · Authors · 2025-11-24

**Comment:**

We thank the reviewers for their comments.

**Withdrawal Confirmation:**

I have read and agree with the venue's withdrawal policy on behalf of myself and my co-authors.